# Effect of Cassava Residue Substituting for Crushed Maize on In Vitro Ruminal Fermentation Characteristics of Dairy Cows at Mid-Lactation

**DOI:** 10.3390/ani10050893

**Published:** 2020-05-20

**Authors:** Yuhui Zheng, Shenglin Xue, Yanyan Zhao, Shengli Li

**Affiliations:** State Key laboratory of Animal Nutrition, College of Animal Science and Technology, China Agricultural University, Beijing 100193, China; zhengyuhui@cau.edu.cn (Y.Z.); xslin20010318@163.com (S.X.); LK20000403@163.com (Y.Z.)

**Keywords:** cassava residue, crushed maize, in vitro gas test, ruminal fermentation characteristics, dairy cows at mid-lactation

## Abstract

**Simple Summary:**

Cassava processing and utilization generates many byproducts, such as cassava residue. On one hand, this residue still contains many nutrients like starch, fiber and minerals. On the other hand, it pollutes the environment if not be utilized properly. Lactating dairy cows need to control the body weight for smooth calving, but too much high energy feed material like maize can influence their lipid metabolism. Cassava residue may be a good option for them since it can be used not only as roughage, but also can provide energy for dairy cows. Therefore, this study was conducted to investigate the effect replacing high energy feedstuff crushed maize with cassava residue on in vitro fermentation characteristics of dairy cows in mid-lactation. This may help provide further in vivo tests with data support, which finally, could alleviate feed shortages, reduce environmental pollution and improve economic benefits of dairy farming.

**Abstract:**

This study was conducted to investigate the effect of using cassava residue to replace crushed maize on in vitro fermentation characteristics of dairy cows at mid-lactation and provide guidance for its utilization. The study included seven treatments with four replicates, which used 0% (control, CON), 5%, 10%, 15%, 20%, 25% and 30% cassava residue to replace crushed maize (air-dried matter basis), respectively. A China-patented automated trace gas recording system was used to perform in vitro gas tests; rumen fluids were collected from three dairy cows at mid-lactation. In vitro dry matter digestibility (IVDMD), gas production (GP), pH, ammonia–N (NH_3_-N) and microbial protein (MCP) content were analyzed after in vitro incubating for 3, 6, 12, 24 and 48 h, respectively; volatile fatty acid (VFA) content was analyzed after in vitro culturing for 48 h. The results showed that with the increase of substitution ratio of cassava residue, the asymptotic gas production (A) increased quadratically (*p* < 0.05), cumulative gas production at 48 h (GP_48_) and the maximum rate of substrate digestion (RmaxS) increased linearly and quadratically (*p* < 0.05), the time at which the maximum gas production rate is reached (TRmaxG) increases linearly (*p* < 0.05). In addition, asymptotic gas production in 30% was significantly higher than the other treatments (*p* < 0.05), RmaxS in 25% and 30% were significantly higher than CON, 5% and 10% (*p* < 0.05). In addition, with the increase of substitution ratio of cassava residue, when in vitro cultured for 6 h and 12 h, NH_3_–N content decreased linearly and quadratically (*p* < 0.05). NH_3_–N content in 30% was significantly lower than the other treatments except 20% and 25% (*p* < 0.05) after cultivating for 6 h. Moreover, the content of iso-butyrate, iso-valerate, valerate and total VFA (tVFA) decreased linearly and quadratically (*p* < 0.05), acetate decreased quadratically (*p* < 0.05) with the increase of substitution ratio of cassava residue. In conclusion, when the cassava residue substitution ratio for crushed maize was 25% or less, there were no negative effects on in vitro ruminal fermentation characteristics of dairy cows at mid-lactation.

## 1. Introduction

Cassava is a staple food for more than 300 million people all over the world [1]. Studies have shown that cultivation of cassava is currently expanding world-wide and it is consumed widely in most regions of South America, Africa and Asia [2]. The processing and utilization of cassava generate a lot of byproducts, such as cassava residue. Unused cassava residue usually rots, pollutes the environment and endangers the health of both human and animals [3]. Research has shown that cassava residue has 74.4% of nitrogen-free extract content, 3519 Kcal/kg crude energy [4] and it is rich in amino acids and minerals such as copper, potassium, manganese and iron [5]. In addition, cassava residue is much cheaper compared with common feedstuffs like crushed maize, so that it is an unconventional feedstuff with great utility value for livestock.

One study showed that diets with 15% cassava bagasse will make milk production increase 13.2% and its operational cost decrease 42.8%, compared to controls [6]. Using cassava residue (less than 20%) to replace maize in the dietary of Hu lambs could linearly improve its growth performance along with antioxygenic property and liver function were also affected [7]. One study indicated that using diet with 32% of cassava residue to feed lactating dairy cows at early lactation, their milk production reduced but the constituents of milk was not affected. Besides, metabolic parameters, glucose and urea nitrogen in plasma were also within appropriate levels [8]. These studies illustrated that cassava residue could be used to replace crushed maize and it can improve the growth performance of the dairy cows at early lactation or lambs. However, the application of replacing crushed maize with cassava residue to feed dairy cows at mid-lactation needs further exploration. Dairy cows at mid-lactation need to control their body weight and prevent them from getting to fat in order to ensure the smooth calving, but too much high energy feedstuff like maize intake could influence their lipid metabolism [9]. Cassava residue has high content of fiber and starch, so that it could be a good option for dairy cows at mid-lactation. It cannot only be used as roughage, but also can be used to provide energy for dairy cows. Therefore, this experiment aims to investigate the effect of cassava residue substituting for crushed maize on ruminal fermentation characteristics of dairy cows at mid-lactation by in vitro gas test. Hope to provide basis for using appropriate ratio of cassava residue to replace crushed maize as a kind of unconventional feedstuff for dairy cows.

## 2. Materials and Methods

The experimental protocols used in this experiment were approved by Institutional Animal Care and Use Committee of China Agricultural University (Beijing, China) (No. AW09089102–1). The experiment was carried out at Zhongdi Animal Husbandry Technology Co., Ltd. (Beijing, China).

### 2.1. Experimental Products

Cassava residue (Produced in Thailand, extracted starch first then pressed physically) was provided by Jiuzheng Biotechnology Co., Ltd. (Beijing, China) and crushed maize was provided by Zhongdi Animal Husbandry Technology Co., Ltd. (Beijing, China). Feed samples were dried in the oven at 65 °C for 48 h to constant weight then ground to pass through a 1-mm screen (40 mesh) before the determination of nutrient contents according to AOAC (2012) [10] and in vitro gas test. Nutrient contents of cassava residue and crushed maize are showed in Table 1.

There were seven treatments with four replicates in this experiment. Crushed maize was replaced by cassava residue at a ratio of 0%, 5%, 10%, 15%, 20%, 25% and 30% (air-dried matter basis). These treatments were diagnosed with CON (control), 5%, 10%, 15%, 20%, 25% and 30%, respectively. Ingredients composition and nutrient content of fermentable substrates in each treatment were detailed in Table 2.

### 2.2. Experimental Materials and Design

The experiment used the China-patented automated trace gas recording system (AGRS-III, Beijing, China) [11] to record the cumulative gas production (GP) in real time. Following the experimental method proposed by Menke [12], buffer was configured, and CO_2_ was injected slowly and continuously for about 30 min before inoculation. The pH of the buffer was adjusted to 6.8 then warmed at 39 °C in water bath. Rumen fluids were collected from three adult Holstein dairy cows (130 ± 20 days in milk) with permanent rumen fistulas before feeding in the morning. The diet of the dairy cows was formulated by CMP software (Moate et al., New York, NY, USA). Net energy of lactation of the diet was 1.75 Mcal/kg and crude protein content of the diet was 19.6%. Dry matter intake of the cows was 24.0 kg/d and they were fed twice daily and had free access to water. The collected rumen fluids were filtered by four layers of gauze and then placed in water bath at 39 °C, which was continuously stirred and poured into CO_2_. Five hundred milligram of substrates, 50 mL of pre-warmed buffer and 25 mL of rumen fluids were accurately transferred to glass bottles (volume capacity of 120 mL). An extra four substrate-free bottles without the addition of substrates were served as blanks, and 50 mL of pre-warmed buffer and 25 mL of filtered rumen fluids were also added to them. All the bottles were sealed with Hungate’s stoppers and screw caps and immediately connected to gas channel inlets of AGRS-III through medical transfusion tubes and needles. All bottles were incubated at 39 °C and the system was shut down after culturing for 3 h, 6 h, 12 h, 24 h and 48 h, respectively. After the incubation, the bottles were disconnected from the system and all materials in the bottles were transferred to pre-dried and weighed nylon bags (300 mesh, 9 cm × 14 cm) to get culture fluids. Then pH was determined in the culture fluids and nylon bags were washed until clarified and were put into the oven at 65 °C to determine the in vitro dry matter disappearance (IVDMD). Culture fluids were used to determine the contents of ammonium nitrogen (NH_3_-N), microbial proteins (MCP) and volatile fatty acids (VFA).

### 2.3. Test Indicators and Methods

#### 2.3.1. In Vitro Dry Matter Digestibility, Gas Production and Kinetic Parameters of Gas Production

IVDMD was calculated by differential subtraction according to the dry matter content of substrates before and after in vitro incubation. The cumulative gas production data were recorded by AGRS-Ш system and fitted to Groot model as Equation (1) [13]:GP_t_ = A/(1 + (C/t)^B^)(1)
where GPt means the cumulative gas production (mL/g DM) at incubation time t (h), A means the asymptotic gas production (mL/g DM), B means a sharpness parameter determining the shape of the curve, C means the time (h) at which half of A is reached and t means in vitro incubation time.

The time at which maximum rate of substrate degradation is reached (TRmaxS, h), the maximum rate of substrate digestion (RmaxS,/h), the time at which RmaxG is reached (TRmaxG, h) and the maximum gas production rate (RmaxG, mL/h) were calculated with A–C as Equations (2)–(5) [14]:TRmaxS = C × (B − 1)^(1/B)^(2)
RmaxS = (B × TRmaxS^(B − 1)^)/(C^B^ + TRmaxS^B^)(3)
TRmaxG = C × ((B − 1)/(B + 1))^(1/B)^(4)
RmaxG = (A × C^B^ × B × TRmaxG^−B − 1^)/(1 + C^B^ × TRmaxG^(−B)^)^2^(5)

#### 2.3.2. pH, Ammonia–N, Microbial Protein and Volatile Fatty Acid Content

The determination of the pH in culture fluids was carried out with Mettler Five Easy Plus series pH meter, the determination of NH_3_–N content was performed by bright blue colorimetry [15], the content of MCP was determined according to the Coomassie bright blue colorimetry [16] and the VFA content was determined by gas chromatography method [17].

### 2.4. Statistical Methods

Data were simply summarized in Excel 2017 (Microsoft, Redmond, WA, USA) and then fitted with SAS 9.2 NLIN (SAS institute, Carry, NC, USA) program to obtain the kinetics parameters of gas generation A, B, C, TRmaxG, RmaxG, TRmaxS and RmaxS. Data were analyzed by one-way ANOVA using the general linear model procedure in SAS 9.2 (SAS institute, Carry, NC, USA) and Tukey method for multiple analysis. Polynomial contrasts were conducted to determine the linear and quadratic effects of different cassava residue substitution ratio for crushed maize. Significance was designated at *p* ≤ 0.05.

## 3. Results

### 3.1. In Vitro Dry Matter Digestibility

As can be seen from Table 3, with the increase of cassava residue substitution ratio, there was no significant difference in IVDMD at each in vitro incubation time (*p* > 0.05).

### 3.2. Gas Production and Kinetic Parameters

It can be obtained from Table 4, with the increase of cassava residue substitution ratio, Kinetic parameters A increased quadratically (*p* < 0.05), GP_48_, C and RmaxS increased linearly and quadratically (*p* < 0.05), TRmaxG increases linearly (*p* < 0.05). In addition, Kinetic parameters A of 30% were significantly higher than the other treatments (*p* < 0.05), GP_48_ of 30% was significantly higher than CON and 10% (*p* < 0.05), C of 30% was significantly lower than the other treatments except 25% (*p* < 0.05), RmaxS in 25% and 30% were significantly higher than CON, 5% and 10% (*p* < 0.05) and TRmaxG in 30% was significantly higher than CON, 5% and 25% (*p* < 0.05), TRmaxS in 20% was significantly longer than the other treatments except 25% and 30% (*p* < 0.05).

### 3.3. Ammonia–N Content

Obtained from Table 5, with the increase of substitution ratio of cassava residue, when in vitro cultured for 6 h and 12 h, NH_3_–N content all decreased linearly and quadratically (*p* < 0.05). At 6 h, NH_3_–N content in 30% was significantly lower than the other treatments except 20% and 25% (*p* < 0.05). In addition, there were no significant differences between treatments at other in vitro incubation time (*p* > 0.05).

### 3.4. Microbial Protein Content

As can be seen from Table 6, with the increase of cassava residue substitution ratio, there was no significant difference in MCP content at each in vitro incubation time (*p* > 0.05).

### 3.5. pH

It can be known from Table 7, when in vitro cultured for 12 h, pH decreased linearly (*p* = 0.05) as the cassava residue substitution ratio increased; when in vitro cultured for 48 h, pH increased both linearly and quadratically (*p* < 0.05). In addition, pH in 30% was significantly higher than CON, 5% and 20% (*p* < 0.05) when incubated for 48 h and there were no significant differences between treatments at other in vitro incubation time (*p* > 0.05).

### 3.6. Volatile Fatty Acid Content

As can be seen from Table 8, with the increase of cassava residue substitution ratio, the content of iso-butyrate, iso-valerate, valerate and tVFA in each treatment decreased linearly and quadratically (*p* < 0.05) and acetate decreased quadratically (*p* < 0.05). The acetate content in 30% was significantly lower than other treatments except 20% (*p* < 0.05) and the iso-butyrate, iso-valerate and valerate content in 30% lower than other treatments except 25% (*p* < 0.05), the tVFA content in 30% was significantly lower than other treatments except 10% and 20% (*p* < 0.05).

## 4. Discussion

Dry matter degradation rate is an important factor that affects the dry matter intake of dairy cows. It is positively correlated with the degradation of protein, amino acids and starch of feed. In addition, rumen microorganisms consume carbohydrates and other nutrients to produce methane, hydrogen, carbon dioxide and other gases. Cumulative gas production is an important indicator reflect not only the utilization degree of the substrate by rumen microorganisms but also the nutritional value of substrate [18,19]. The amount of gas produced by in vitro fermentation is closely related to the degradation rate of carbohydrates in the substrate. The larger the amount of gas produced, the better the fermentation degree of the substrate—and the higher the rumen digestion degree will be. Current research indicates that in vitro gas production is negatively correlated with the content of NDF in the substrates [20]. Cassava residue contains much NDF and ADF that are not easily fermented. However, in this experiment, with the increase of cassava residue substitution ratio, there was no significant difference in IVDMD at each in vitro incubation time, but cumulative gas production increased linearly when in vitro cultured for 48 h. Meanwhile, this result was not consistent with the previous study which showed that gas production was positively correlated to ruminal DM digestibility [12]. Study dedicated that if the ratio of carbon and nitrogen of the substrate is more suitable for rumen microorganisms, the utilization of the substrates will increase [21]. Moreover, the study also showed that if the ratio of fermentable nutrients and unfermentable structural carbohydrates is more reasonable, it could be more easily for microorganisms to utilize [22]. Therefore, the results of this experiment may be that when the two feed ingredients were mixed, the ratio of fermentable nutrients and unfermentable structural carbohydrates became more suitable than single feed, so that the ratio of carbon and nitrogen of the substrate is more suitable for the fermentation of rumen microorganisms. As a result, although IVDMD showed no significant differences between each treatment, cumulative gas production increased linearly with the increase of cassava residue substitution ratio. In order to further study the mechanism of this phenomenon, we will focus on the effect of using cassava residue to replace crushed maize on the rumen microflora composition and microbial activity of dairy cows at mid-lactation in latter experiments.

NH_3_–N is an important metabolite of nitrogen in the rumen, and is also the main nitrogen source for MCP synthesis. The variation of NH_3_–N content is an important indicator to evaluate the balance between the utilization of nitrogen in the diet by rumen microorganisms and the synthesis of MCP [23]. Studies showed that the content of NH_3_–N in rumen fluids is negatively correlated with the content of carbohydrate in the substrate. The increase of carbohydrate content will make rumen microorganisms become more active, so that their absorption of nitrogen will be increased. Under this circumstance, the content of NH_3_–N in culture fluids will go down [20]. In this experiment, NH_3_–N content all decreased quadratically and linearly with the increase substitution ratio of cassava residue when in vitro cultured for 6 h and 12 h. This may be related to the fact that with the increase of substitution ratio of cassava residue, the ratio of energy and nitrogen in substrates is not conducive to both growth of rumen microorganisms and decomposition of nitrogen-containing substrates. This is consistent with the viewpoint proposed by Kand et al. in 2018, which indicated that rumen microorganisms have dependence, selectivity and timeliness on nitrogen-containing materials, and they are the most direct indicator of the nitrogen synchronization [24].

MCP provides 60%–70% of protein requirements for ruminants and is the main nitrogen source for ruminants [25]. It also can reflect both efficiency in using nitrogen of rumen microorganism and the number of rumen microflora [26]. Studies indicated that the synthesis of MCP mainly depends on the utilization efficiency of carbohydrates and proteins in rumen. When the release of ammonia and energy in rumen is not synchronous, the utilization rate of substrates will decrease, which leads to a decrease in MCP synthesis [27,28,29]. In addition, the synthesis of MCP requires both energy and carbon scaffolding [30]. Studies showed that diet with greater concentration of nonstructural carbohydrates could increase the utilization of ruminal NH_3_–N for microbial protein synthesis of dairy cows [31]. Only when the release rate of ammonia is synchronized with energy and carbon scaffolding, the nitrogen fixation effect of microorganisms will be optimal [28]. Therefore, in this experiment, with the increase of in vitro incubation time, MCP content in each treatment increased first and then decreased, which may be related to the synchronous change of nitrogen release, energy and carbon in the substrate with the extension of time.

The pH of rumen fluids is a comprehensive indicator of rumen fermentation level, which is influenced by many factors such as diet properties, saliva secretion, osmotic pressure, contents of volatile fatty acids and other organic acids in rumen, rumen water flow and buffering power of feed. Cows maintain rumen pH at 5.5–7.5 by a complex acid–base regulation system [32]. In this experiment, the pH in all treatments was within the normal range, which indicated that when the substitution ratio of cassava residue for crushed maize was within 30%, it could provide stable and suitable growth environment for rumen microorganisms of dairy cows at mid-lactation.

Volatile fatty acids are important products of rumen microbial fermentation and could provide 70%–80% of energy for ruminants [33]. Acetate is a major precursor of ruminant animal fat synthesis and it can be used to synthesis short chain fatty acids in milk fat in mammary gland [34]. Propionate is an important precursor of glucose, which is mainly used in body fat and lactose synthesis. Propionate producing can competitively consume hydrogen and inhibit methane synthesis [32]. The ratio of acetate to propionate reflects the energy utilization of ruminants and feed fermentation mode and it should be bigger than 2.2 [35]. The production rate of propionate is proportional to the deposition of nitrogen in feed, and the mismatch of acetate and propionate yield will affect the deposition of nitrogen [36]. In this experiment, with the increase of cassava residue substitution ratio after in vitro culturing for 48 h, the contents of iso-butyrate, iso-valerate, valerate and total VFA in each treatment decreased linearly and quadratically, while the contents of acetate in each treatment decreased quadratically. This indicates that when the substitution ratio of cassava residue is 30%, the synthesis of VFA in rumen of dairy cows will be adversely affected. Studies showed that valerate, iso-butyrate and iso-valerate are microbial fermentation products of valine, leucine and isoleucine in the protein of feed, respectively [37]. Moreover, iso-butyrate and iso-valerate belong to branched-chain VFA, which can stimulate the activity of crude fiber decomposing bacteria, increase the biomass of structural carbohydrate decomposing bacteria and improve the digestibility of dry matter [38]. In this experiment, the yield of these three VFA in treatment 30% was the lowest in each treatment, that is, when the substitution ratio of cassava residue was 30%, the metabolism of these three amino acids in substrate would be inhibited by rumen microorganisms.

## 5. Conclusions

The results of this in vitro experiment showed that when the ratio of cassava residue to substitute crushed maize was 25% or less, there was no adverse effect on rumen fermentation characteristics of dairy cows at mid-lactation. Further tests may be conducted to investigate the effect of using cassava residue substituting for crushed maize on rumen microflora composition and microbial activity of dairy cows at mid-lactation.

## Figures and Tables

**Table 1 animals-10-00893-t001:** Nutrient contents of cassava residue and crushed maize (air-dried matter basis, %).

Items	DM	NDF	ADF	CP	EE	Ca	P	Ash	Starch
Crushed maize	96.24	8.05	2.56	8.30	2.23	0.15	0.19	1.71	71.30
Cassava residue	96.51	30.54	23.09	8.46	0.05	0.91	0.21	5.79	49.00

DM: dry matter; NDF: neutral detergent fiber; ADF: acid detergent fiber; CP: crude protein; EE: ether extract; Ca: calcium; P: phosphorus. All values were analyzed.

**Table 2 animals-10-00893-t002:** Ingredients composition and nutrient content of fermentable substrates in each treatment (air-dried matter basis, %).

Items	Treatments						
	CON	5%	10%	15%	20%	25%	30%
Ingredients							
Crushed maize	100.00	95.00	90.00	85.00	80.00	75.00	70.00
Cassava residue	0.00	5.00	10.00	15.00	20.00	25.00	30.00
Nutrient levels							
DM	96.24	96.25	96.27	96.28	96.29	96.31	96.32
NDF	8.05	9.17	10.30	11.42	12.55	13.67	14.80
ADF	2.56	3.59	4.61	5.64	6.67	7.69	8.72
CP	8.30	8.31	8.32	8.32	8.33	8.34	8.35
EE	2.23	2.12	2.01	1.90	1.80	1.69	1.58
Ca	0.15	0.19	0.23	0.26	0.30	0.34	0.38
P	0.19	0.19	0.19	0.19	0.19	0.20	0.20
Ash	1.71	1.91	2.12	2.32	2.53	2.73	2.93
Starch	71.30	70.19	69.07	67.96	66.84	65.73	64.61

CON (control), 5%, 10%, 15%, 20%, 25% and 30%: using 0%, 5%, 10%, 15%, 20%, 25% and 30% cassava residue to replace crushed maize (air-dried matter basis), respectively; DM: dry matter; NDF: neutral detergent fiber; ADF: acid detergent fiber; CP: crude protein; EE: ether extract; Ca: calcium; P: phosphorus. Values in CON were analyzed and other values were calculated.

**Table 3 animals-10-00893-t003:** Effects of different cassava residue substitution ratio for crushed maize on in vitro IVDMD in dairy cows at mid-lactation (%).

In Vitro Incubation Time	Treatments							SEM	*p*-Value		
	CON	5%	10%	15%	20%	25%	30%		Treatment	Linear	Quadratic
3 h	39.54	39.85	37.85	37.12	40.66	40.88	40.06	0.582	0.82	0.43	0.47
6 h	45.68	46.36	46.48	46.60	49.12	48.65	44.77	0.598	0.79	0.68	0.38
12 h	77.97	81.09	80.09	78.62	78.94	78.80	80.09	0.418	0.47	0.89	0.99
24 h	89.09	89.34	89.68	87.95	86.54	89.68	86.97	0.378	0.15	0.22	0.47
48 h	91.32	93.01	90.55	91.85	90.58	90.57	90.91	0.526	0.56	0.32	0.61

CON (control), 5%, 10%, 15%, 20%, 25% and 30%: using 0%, 5%, 10%, 15%, 20%, 25% and 30% cassava residue to replace crushed maize (air-dried matter basis), respectively; IVDMD: in vitro dry matter digestibility; SEM: standard error of the mean.

**Table 4 animals-10-00893-t004:** Effects of different cassava residue substitution ratio for crushed maize on in vitro gas production and kinetic parameters in dairy cows at mid-lactation.

Items	Treatments							SEM	*p*-Value		
	CON	5%	10%	15%	20%	25%	30%		Treatment	Linear	Quadratic
GP_48_ (mL/g)	178.25 ^b^	223.19 ^a,b^	178.78 ^b^	227.16 ^ab^	224.43 ^a,b^	243.88 ^a,b^	257.96 ^a^	7.671	<0.01	<0.01	<0.01
A (mL)	233.99 ^b,c,d^	267.05 ^b^	184.12^e^	246.64 ^b,c^	211.51 ^d,e^	219.88 ^c,d^	311.27 ^a^	8.817	<0.01	0.19	<0.01
B (h)	1.16	1.19	1.29	1.44	1.38	1.17	1.18	0.038	0.49	0.86	0.17
C (h)	7.13 ^a^	6.02 ^b,c^	6.08 ^b^	5.46 ^b,c^	5.73 ^b,c^	5.07 ^c,d^	4.65 ^d^	0.170	<0.01	<0.01	<0.01
TRmaxG (h)	0.45 ^b^	0.79 ^b^	1.02 ^a,b^	1.32 ^a,b^	0.94 ^a,b^	0.81 ^b^	1.99 ^a^	0.117	0.02	0.02	0.06
RmaxG (h)	16.27 ^e^	35.32 ^a^	20.01 ^d,e^	32.23 ^a,b^	25.39 ^c,d^	35.13 ^a^	28.48 ^b,c^	1.577	<0.01	0.06	0.09
TRmaxS (h)	1.13 ^b^	0.89 ^b^	1.05 ^b^	1.14 ^b^	2.21 ^a^	1.55 ^a,b^	1.54 ^a,b^	0.164	<0.01	0.36	0.20
RmaxS (mL/h)	0.13 ^c^	0.14 ^b,c^	0.13 ^c^	0.17 ^a^	0.16 ^a,b^	0.17 ^a^	0.17 ^a^	0.005	0.01	<0.01	<0.01

CON (control), 5%, 10%, 15%, 20%, 25% and 30%: using 0%, 5%, 10%, 15%, 20%, 25% and 30% cassava residue to replace crushed maize (air-dried matter basis), respectively; SEM: standard error of the mean; GPt: the cumulative gas production (mL/g DM) at incubation time t (h); A: the asymptotic gas production (mL/g DM); B: a sharpness parameter determining the shape of the curve; C: the time (h) at which half of A is reached and t is in vitro incubation time; TRmaxS: The time at which maximum rate of substrate degradation is reached (h); RmaxS: the maximum rate of substrate digestion (/h); TRmaxG: the time at which RmaxG is reached (h); RmaxG: the maximum gas production rate (mL/h). ^a,b,c,d,e^: within a row with different superscripts differ significantly (*p* ≤ 0.05).

**Table 5 animals-10-00893-t005:** Effects of different cassava residue substitution ratio for crushed maize on in vitro NH_3_–N content in dairy cows at mid-lactation (mg/dL) ^1^.

In Vitro Incubation Time	Treatments							SEM	*p*-Value		
	CON	5%	10%	15%	20%	25%	30%		Treatment	Linear	Quadratic
3 h	13.35	12.19	13.23	13.07	12.62	12.35	12.10	0.248	0.78	0.29	0.54
6 h	11.58 ^a^	11.15 ^a,b^	10.91 ^a,b^	10.86 ^a,b^	9.10 ^b,c^	10.49 ^a,b,c^	8.61 ^c^	0.297	0.03	<0.01	<0.01
12 h	7.21	6.94	5.81	6.23	5.12	5.42	6.26	0.211	0.08	0.05	0.02
24 h	16.51	16.80	16.71	16.57	15.40	16.17	16.71	0.664	0.99	0.84	0.96
48 h	39.61	45.11	40.70	46.13	41.97	46.41	45.66	1.114	0.52	0.18	0.40

^1^ CON (control), 5%, 10%, 15%, 20%, 25% and 30%: using 0%, 5%, 10%, 15%, 20%, 25% and 30% cassava residue to replace crushed maize (air-dried matter basis), respectively; NH_3_-N: ammonia-N; SEM: standard error of the mean. ^a,b,c^: within a row with different superscripts differ significantly (*p* ≤ 0.05).

**Table 6 animals-10-00893-t006:** Effects of different cassava residue substitution ratio for crushed maize on in vitro MCP content in dairy cows at mid-lactation (μg/mL).

In Vitro Incubation Time	Treatments							SEM	*p*-Value		
	CON	5%	10%	15%	20%	25%	30%		Treatment	Linear	Quadratic
3 h	235.03	218.91	225.93	227.92	235.30	224.31	227.53	1.711	0.16	0.95	0.97
6 h	248.59	245.52	246.93	250.89	248.13	250.47	245.85	1.409	0.94	0.87	0.84
12 h	273.72	278.41	276.46	281.76	277.56	283.17	260.06	6.500	0.98	0.75	0.72
24 h	257.43	266.08	255.95	270.79	243.76	264.63	255.46	4.140	0.77	0.80	0.93
48 h	168.74	170.15	176.97	164.63	174.53	175.90	185.34	2.282	0.36	0.10	0.16

CON (control), 5%, 10%, 15%, 20%, 25% and 30%: using 0%, 5%, 10%, 15%, 20%, 25% and 30% cassava residue to replace crushed maize (air-dried matter basis), respectively; MCP: microbial protein; SEM: standard error of the mean.

**Table 7 animals-10-00893-t007:** Effects of different cassava residue substitution ratio for crushed maize on in vitro pH in dairy cows at mid-lactation.

In Vitro Incubation Time	Treatments							SEM	*p*-Value		
	CON	5%	10%	15%	20%	25%	30%		Treatment	Linear	Quadratic
3 h	7.44	7.43	7.43	7.43	7.40	7.42	7.42	0.006	0.56	0.16	0.27
6 h	7.47	7.47	7.45	7.45	7.49	7.41	7.40	0.013	0.53	0.12	0.18
12 h	7.20	7.21	7.21	7.21	7.18	7.14	7.19	0.008	0.13	0.05	0.15
24 h	7.10	7.13	7.13	7.13	7.15	7.14	7.14	0.008	0.81	0.15	0.26
48 h	6.26 ^b^	6.23 ^b^	6.29 ^a,b^	6.29 ^a,b^	6.27 ^b^	6.30 ^a,b^	6.34 ^a^	0.009	0.04	<0.01	0.02

CON (control), 5%, 10%, 15%, 20%, 25% and 30%: using 0%, 5%, 10%, 15%, 20%, 25% and 30% cassava residue to replace crushed maize (air-dried matter basis), respectively; SEM: standard error of the mean. ^a,b^: within a row with different superscripts differ significantly (*p* ≤ 0.05).

**Table 8 animals-10-00893-t008:** Effects of different cassava residue substitution ratio for crushed maize on in vitro VFA concentration after cultivating for 48 h in dairy cows at mid-lactation (mmol/L).

Items	Treatments							SEM	*p*-Value		
	CON	5%	10%	15%	20%	25%	30%		Treatment	Linear	Quadratic
Acetate	1.02 ^a^	1.06 ^a^	1.02 ^a^	1.10 ^a^	1.01 ^a,b^	1.08 ^a^	0.91 ^b^	0.012	<0.01	0.08	<0.01
Propionate	0.41	0.42	0.42	0.43	0.39	0.43	0.41	0.005	0.47	0.91	0.76
Iso-butyrate	0.52 ^a^	0.49 ^a^	0.45 ^b^	0.46 ^b^	0.42 ^b^	0.39 ^b,c^	0.36 ^c^	0.014	0.02	<0.01	<0.01
Butyrate	0.81	0.80	0.78	0.82	0.79	0.76	0.73	0.011	0.53	0.08	0.13
Iso-valerate	0.97 ^a^	0.88 ^a,b^	0.89 ^a,b^	0.91 ^a,b^	0.86 ^a,b^	0.81 ^b,c^	0.75 ^c^	0.015	<0.01	<0.01	<0.01
Valerate	0.51 ^a^	0.48 ^a^	0.49 ^a^	0.52 ^a^	0.49 ^a^	0.43 ^a,b^	0.41 ^b^	0.009	0.02	<0.01	<0.01
Acetate/Propionate	2.53	2.50	2.47	2.56	2.65	2.38	2.32	0.033	0.29	0.34	0.24
tVFA	4.28 ^a^	4.17 ^a^	4.05 ^a,b^	4.19 ^a^	3.97 ^a,b^	4.22 ^a^	3.56 ^b^	0.049	<0.01	<0.01	<0.01

CON (control), 5%, 10%, 15%, 20%, 25% and 30%: using 0%, 5%, 10%, 15%, 20%, 25% and 30% cassava residue to replace crushed maize (air-dried matter basis), respectively; VFA: volatile fatty acids; tVFA: total volatile fatty acids; SEM: standard error of the mean. ^a,b,c^: Means within a row with different superscripts differ significantly (*p* ≤ 0.05).

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
