# Peer review of "Effect of Cassava Residue Substituting for Crushed Maize on In Vitro Ruminal Fermentation Characteristics of Dairy Cows at Mid-Lactation"

_animals, 2020, doi:10.3390/ani10050893_

Round 1

Reviewer 1 Report

Lines 25 – 28: The phrase: “The result showed that IVDMD decreased linearly when in vitro cultured for 24 h 26 and 48 h (p < 0.05)” is incomplete.

It is mentioned, in lines 140 – 141, that, dry mater in vitro digestibility decreases linearly with the increment of cassava in the diet; however, this is not what the data shows

Tables 3, 7 and 8 are confusing; in the column SEM it is shown P<0.01, and in the column P-value: 0.19. Please check and correct.

Table 4. Please indicate the meaning of the abbreviations. Also please indicate the method used to test the mean differences. The A parameter is not shown.

Lines 148 and 150 need to be revised, there is no agreement to what is in the Table 4 (expected? excepted?). Several descriptions do not agree with the Table 4. For instance: you say that  “ TRmaxS in group 25% and  30% was significantly longer than the other groups (p < 0.05). “, but in the table 4 most of the means (except of the 20%) for TRmaxS  are followed by the literal “a” which indicates that they are similar.

Lines 161-162 are confusing

Line 164: You say: “At 6 h, NH3-N content in group 30% was significantly lower than the other groups expect group 20 (p <  0.05)”; please, check in the Table 5, also 25% is similar to 20 and 30%.

Lines 168-170 and Table 6: In the text, you indicate that the data in table 6 was compared within (3, 6 12…h ) and between columns (0, 5 10..%). Please indicate the type of design (a latin square?) or factor arrangement (factorial?) used to be able to make this type of comparison.

Line 174 – 177. Confusing, please check and correct.

Line 181 – 186. Needs correction on redaction.

In the discussion (Lines 191 – 200) you argued that there is a linear reduction on IVDMD. Data in Table 3, for the 24 and 48 h, is very similar among levels of cassava (max: 0.89, min: 0.87 (for 20 and 30% level), this is only 0.03!!!! different. Even more your P-value for both (24 and 48 h) indicate no difference, but the lineal regression have p-value lower that 0.05. Please revised your regression analysis it seems to be wrong.

The arguments in the lines 200- 2004 do not agree with the data in tables 3 and 4. Total gas production tended to increase as the cassava increment in the diet, while IVDMD goes in the other way. The argument show in the section 207 – 209, dos not neccesarily explain this situation.

Please revise this sentence (216-217): “The higher the content of starch in the substrate, the lower the degradation rate of protein” We have found that energy supplementation make microganismos more active and therefore rate of degradation is increased except if the protein is protected somehow (with taninns, etc) to avoid rumen degradation.

Line 223 Please add the year

Line 231 sync?.

Author Response

Respective editor, thank you very much for your valuable comments. We have revised our manuscript thoroughly. Please see the attachment.

Reviewer 2 Report

The manuscript submitted for review focuses on the evaluation of the effect of Cassava residue substituting for crushed maize on rumen fermentation characteristics of dairy cows. Zheng and co-authors described different combinations of cassava residue that are replaced by crushed maize to be used in dairy cow diets. They specifically focused on the use of the in vitro gas production method to compare both substrates in terms of digestibility, kinetic of fermentation, microbial protein content and fermentation profile. The topic of the manuscript fall within the general scope of the Journal and it is based on a routinely accepted experimental technique to evaluate the nutritional value of ruminant feeds.

While there are several redeeming features of the manuscript that required large amounts of time and effort to complete, there are also several areas of concern. I have been able to observe, without being a native English speaker, that this manuscript has a number of general and specific grammar and language issues that need to be addressed. 

Some examples are indicated below:

L30 : …linearly (p < 0.05).D

L57: ...affected [7].A ...

L67: : Is suggested to replace …feed stuff … by ... feedstuff  

L87: L87 through L108: Please revise grammar and language use.

L135: Please replace "Turkey ..." by "Tukey..."

L216: The higher the content of...

A general review of the manuscript was made, based mainly on methodological aspects related to the use of the in vitro techniques on the evaluation of the effect of dietary treatments and rumen microbial fermentation. On this respect, the main general observations are specifically addressed below.

Introduction section:

Authors state in L60-61 that "... the application of cassava residue in dairy cows needs further exploration...". Only as an exercise, I just did a quick search on internet, using "cassava residue" and "dairy cows" and "cassava residue" as a key words and too many scientific paper were found.

Mat&Met section:

  • Treatments: For a better understanding of the experimental procedure, it is suggested replacing group by treatments in text and tables. Maybe 0% treatment will be called control.
  • The study was carried out without replication (a single experimental period) and no blank was used for correction and/or this aspect is not mentioned in the paper.
  • Although the in vitro gas test is a widely accepted method, observed results are not directly applicable to real production conditions. Authors should be cautious when making global recommendations (i.e. as in the last sentence of the simple summary).
  • Authors did not mention nothing about adjustment of the model selected to predict kinetic parameters (e.g. Chart plotting or adjustment parameters).

Results and discussion sections:

L140-141: Authors describe a significant linear decrease (p<0.05), but P value of ANOVA indicated in Table 3 was not significant. If the ANOVA test is not significant, multiple comparison test will not be performed.

Once an ANOVA test has been completed, the researcher may still need to understand sub group differences among the different experimental and control groups. If ANOVA has found significant, then a multiple comparison test will be performed.

Authors pointed out a linear decrease in IVDMD (discussion section), which is not supported by the ANOVA results. 

Authors are encouraged to make a comprehensive review of grammar and writing form of these sections to make it easy to read or to understand what is written. 

Author Response

(The authors gave the same response as above.)
